# Feasibility of Neovessel Embolization in a Large Animal Model of Tendinopathy: Safety and Efficacy of Various Embolization Agents

**DOI:** 10.3390/jpm12091530

**Published:** 2022-09-18

**Authors:** Julien Ghelfi, Ian Soulairol, Olivier Stephanov, Marylène Bacle, Hélène de Forges, Noelia Sanchez-Ballester, Gilbert Ferretti, Jean-Paul Beregi, Julien Frandon

**Affiliations:** 1Faculty of Medicine, University of Grenoble Alpes, 38000 Grenoble, France; 2Department of Radiology, Grenoble-Alpes University Hospital, 38000 Grenoble, France; 3Department of Pharmacy, Nîmes University Hospital, 30900 Nîmes, France; 4ICGM, University of Montpellier, CNRS, ENSCM, 34090 Montpellier, France; 5Anatomopathology Department, Grenoble University Hospital, 38000 Grenoble, France; 6Faculty of Medicine, Montpellier Nîmes University, RAM-PTNIM, 30900 Nîmes, France; 7Department of Medical Imaging, Nîmes University Hospital, Imagine UR UM 103, University of Montpellier, 34090 Montpellier, France

**Keywords:** tendinopathy, neovessels, embolization, imipenem/cilastatin

## Abstract

Targeting neovessels in chronic tendinopathies has emerged as a new therapeutic approach and several embolization agents have been reported. The aim of this study was to investigate the feasibility of embolization with different agents in a porcine model of patellar tendinopathy and evaluate their safety and efficacy. Eight 3-month-old male piglets underwent percutaneous injection of collagenase type I to induce patellar tendinopathies (*n* = 16 tendons). They were divided into four groups (2 piglets, 4 tendons/group): the control group, 50–100 µm microspheres group, 100–300 µm microspheres group, and the Imipenem/Cilastatin (IMP/CS) group. Angiography and embolization were performed for each patellar tendon on day 7 (D7). The neovessels were evaluated visually with an angiography on day 14. The pathological analysis assessed the efficacy (Bonar score, number of neovessels/mm^2^) and safety (off-target persistent cutaneous ischemic modifications and presence of off-target embolization agents). The technical success was 92%, with a failed embolization for one tendon due to an arterial dissection. Neoangiogenesis was significantly less important in the embolized groups compared to the control group angiographies (*p* = 0.04) but not with respect to histology (Bonar score *p* = 0.15, neovessels *p* = 0.07). Off-target cutaneous embolization was more frequently depicted in the histology of the 50–100 µm microspheres group (*p* = 0.02). Embolization of this animal model with induced patellar tendinopathy was technically feasible with different agents and allowed assessing the safety and efficacy of neovessel destruction. Particles smaller than 100 µm seemed to be associated with more complications.

## 1. Introduction

Tendinopathy is a frequent pathology in the general population and especially in athletes. The most significant symptom is pain; analgesic treatments include physical therapy and icing, oral analgesics, peri-tendinous corticosteroid, or intra-tendinous platelet-rich plasma injections [1,2]. They are generally effective, but up to 10% of patients report chronic pain, with an impact on the patient’s quality of life and impairment of daily activities. The therapeutic alternative is then surgery, which remains invasive with no guaranteed results.

Chronic tendinopathy is accompanied by neoangiogenesis and neoinnervation [3]. A similar mechanism has been reported in osteoarthritis where it was shown to be a cause of chronic pain resistant to the usual analgesics, as it involves neuropathic signaling pathways [4].

Destruction of these pathological neovessels may help break this neurogenic pain cycle. Ultrasound-controlled injections of sclerosing agents targeting the neovessel area have been reported in patients with patellar tendinopathy, with improvement in some patients [5]. Recently, Okuno et al. [6] have shown that arterial embolization by an endovascular approach in osteoarticular pathologies and tendinopathy, in particular, could be an alternative analgesic therapy by targeting these neovessels [7]. To date, several embolization agents of different sizes have been described [7,8,9] but no large randomized studies of the embolization technique in osteoarticular pathologies have yet been reported. Among them, an antibiotic that precipitates in the form of crystals, imipenem/cilastatin (IMP/CS), has been reported with very good analgesic efficacy in patients with tendinopathy or knee osteoarthritis [10,11,12,13,14]. Because of its resorbing nature and small size, it allows for the destruction of neovessels without associated complications. Similar studies with definitive microspheres (MS) have shown less conclusive results, with complications including off-target embolization and skin modifications [15].

The identification of the most adequate embolization agents requires the use of animal models that allow arterial catheterization. However, there are only a few large models of tendinopathy allowing angiographic exploration for embolization. A porcine model of patellar tendinopathy with neoangiogenesis was thus developed by our team [16]. A previous study showed the feasibility and reproducibility of induced patellar tendinopathy after collagenase injection. Neovascularization was confirmed by angiographic findings and pathological analyses.

The main objective of the present study was to assess the feasibility of embolization in this model with various embolization agents and to evaluate their safety and efficacy on neovessel destruction via angiography and histology.

## 2. Materials and Methods

### 2.1. Patellar Tendinopathy Induction

Eight 3-month-old male piglets underwent percutaneous injection of collagenase type I (Sigma-Aldrich, St. Louis, MO, USA) under ultrasound guidance at a dose of 25 mg, according to the model and protocol previously published [16]. It was performed in accordance with the National Institute of Health guidelines for the use of laboratory animals. Authorization of the local government animal rights protection authorities (Languedoc-Roussillon No 36, ID number Nr 2018011916269335 #13156 v3) was obtained.

### 2.2. Angiography Explorations

Endovascular explorations were performed 7 days and 14 days after the collagenase injection on a Fluorostar III (GE Healthcare) following the same protocol as previously published [16]. Briefly, a left carotid arterial access was performed under ultrasound guidance using the Seldinger technique. The femoral artery was catheterized with a 4Fr catheter and the genicular arteries with a 2.0 microcatheter. Injections were performed manually with a 5 mL syringe on DSA imaging. Neovascularization was graded as none, mild, or important by two radiologists in consensus, blinded to the treatment group, in a random order, three months after the procedure.

### 2.3. Embolization Procedure

The piglets were divided into four groups, with four patellar tendons per group (Figure 1):The control group (2 pigs, 4 tendons) underwent diagnostic angiography alone at D7 (no embolization) and D14.Two groups (4 pigs, 8 tendons, 4 tendons in each group) were embolized with calibrated Embosphere^®^ microspheres (Merit Medical, Paris, France) of either 50 to 100 µm (50–100 µm group) or 100 to 300 µm (100–300 µm group) diameter at D7 after collagenase injection. The microspheres were diluted to the 1/20th in NaCl and iodinated contrast was injected 0.1 per 0.1 mL according to the “pruning” technique [17] (Figure 2).The fourth group (2 pigs, 4 tendons) was embolized at D7 using an emulsion of imipenem/cilastatin (IMP/CS group) (500/500 mg) diluted in 10 mL of Visipaque iodinated contrast (GE Healthcare, Marlborough, MA, USA). The mixture was injected 0.1 per 0.1 mL until complete stasis of the feeding artery according to the Martinez et al. technique [18] was achieved (Figure 2).

### 2.4. Histological Analyses

The analyses were performed following the same protocol as previously published [16]. Briefly, after surgical dissection and sacrifice, tendons and skin samples were taken from the outer side of the thigh, near the tendon, in the area where a livedo was reported in piglets. They were fixed in 4% formaldehyde and embedded in paraffin. A pathologist, blinded to the groups, performed the examination on an Olympus BX51 microscope, with a Bonar score [19,20,21] characterization focused on the vascularity subclass index. The number of neovessels/mm^2^ and the presence of persistent embolization agents in the tendons and the skin were also recorded.

### 2.5. In Vitro Analysis

A focus on the microscopy study of the crystallization and solubility of IMP/CS was also carried out. Particle size distribution was microscopically evaluated in three samples of 50 mg IMP/CS (Arrow, Lyon, France) suspensions with a mixture of Visipaque iodinated contrast (GE Healthcare, Marlborough, MA, USA) and NaCl using a volume of 1000 μL and a vortex mixing time of 10 s (Scientific Industries SI™ Vortex-Genie™ 2, Fisher Bioblock Scientific).

The longest diameter of the IMP/CS particles was manually measured using a digital microscope (Keyence VHX-700 Digital Microscope) with an image analyzing software (VHX Analyzer; Itasca, IL, USA). The solubility of IMP/CS in formaldehyde was tested with 10 mg of IMP/CS in 10 mL of formaldehyde and stirred for 10 min. The suspension was then filtered through a 0.45 μm syringe filter (Millipore Milliflex^®^-HV) and UV–Vis spectrophotometrically analyzed in the wavelength range of 200–400 nm using a Specord 200 Plus UV/Vis spectrophotometer (Analytik Jena, Jena, Germany).

### 2.6. Study Endpoints

The primary endpoint was the technical success of the embolization procedure defined as efficient catheterization and embolization of the patellar artery with no neovessel opacification after embolization, as assessed on angiography imaging just after the procedure and compared to the baseline angiography.

Secondary endpoints were the safety and efficacy of various embolization agents. Efficacy was evaluated on neovessel destruction via angiography and histology on day 14, 7 days after embolization. The angiographic visual evaluation compared neovascularization at day 14 versus controls (none, mild, and severe). In regards to the histology, the evaluation of efficacy compared the number of neovessels/mm^2^ and the Bonar score of embolized groups with the control group. Safety was assessed clinically with a behavior scale previously reported [16], weight intake, and screening of off-target embolization using post-embolization immediate clinical cutaneous evaluation (livedo) and delayed histological analysis of the skin around the patellar tendon (inflammatory modifications and persistent embolization agents in the skin). The last objective was to describe the size distribution of the IMP/CS particles as close as possible to the conditions of use as described in the literature in clinical practice [12] (50 mg of IMP/CS and 1 mL of iodinated contrast) as described above.

### 2.7. Statistical Analyses

Quantitative variables are presented using medians and interquartile ranges (1st–3rd) and qualitative variables with numbers and percentages. Quantitative values were compared using the Kruskal–Wallis test when comparing the four groups and the Mann–Whitney test when comparing two groups. Qualitative values were compared using Pearson’s Chi-squared test. Tests were considered significant at the *p* < 0.05 level. Analyses were performed on Excel^®^ (version 16.49) and using Graph Pad Prism version 9.3 (GraphPad Software, San Diego, CA, USA).

## 3. Results

### 3.1. Technical Success

Embolization of the neovessels was successfully achieved in 11/12 patellar tendons (92%). One procedure of the 50–100 µm microspheres group failed because of femoral artery dissection without any clinical consequence.

### 3.2. Safety

No clinical sign of patellar tendinopathy was reported after collagenase injection in all eight piglets. They all reported a walking score of 3/3. No weight loss was reported; the median weight gain between baseline and day 14 was 12% {IQR: 9.3–16}. After embolization, immediate transient livedo was recorded in 7/11 knees (64%), with no difference according to the embolization agent used (*p* = 0.76). On pathological analysis, persistent cutaneous ischemic modifications were more frequently reported in the 50–100 µm microspheres group than in the two other groups (*p* = 0.02) (Table 1). Among the four knees with persistent skin modifications at pathology, persistent embolic agents were depicted in two skin samples (one piglet of the 50–100 µm microspheres group and one piglet of the IMP/CS group) (Figure 3).

### 3.3. Efficacy

Neovascularization was similar between the four groups on the baseline angiography (day 7 after collagenase injection, just before embolization) (Table 2). On day 7 after embolization (day 14 after collagenase injection), it was significantly higher in the control group compared to the groups treated with embolization (microspheres groups or IMP/CS group, *p* = 0.04).

There was no difference in the Bonar score (*p* = 0.15) or the Bonar vascular subclass (*p* = 0.52) between the controls and embolized tendons. The number of neovessels/mm^2^ was lower in the embolized groups compared to the control group, although it was not found significant (*p* = 0.07), especially in the 50–100 µm microspheres group (17 neovessels/mm^2^ {IQR: 16.5–20.5}) as compared to the control group (33 neovessels/mm^2^ {IQR: 29.5–36.8}, *p* = 0.06).

### 3.4. In Vitro Analysis

Over 1500 particles were measured in each condition of the IMP/CS dilution with contrast. More than 96.8% of IMP/CS particles were <40 μm in all conditions tested (Figure 4). For the solubility of IMP/CS in formaldehyde, no absorption peaks were observed, demonstrating the insoluble character of the IMP/CS at the concentration tested.

## 4. Discussion

Embolization of neovessels in this large animal patellar tendinopathy model was feasible with three different embolization agents. Our results showed the efficacy of the technique in terms of neovessel destruction. Embolization was safer with microspheres larger than 100 µm; with smaller embolization agents, microspheres, or IMP/CS reporting off-target embolization inducing cutaneous ischemia.

### 4.1. Neoangiogenesis

Our results showed the efficacy of embolization with significant destruction of neovessels in embolized piglets as compared to controls. They are in accordance with a rat model of frozen shoulder treated by embolization with IMP/CS [22]. In their study, a decreased number of mononuclear inflammatory cells was also described, suggesting that transcatheter arterial embolization may also improve the inflammatory reaction in frozen shoulder. Recently, a ram model of knee osteoarthritis was evaluated by intra-arterial injection of mono-iodoacetate. Despite the limited number of treated knees (75 µm calibrated microparticles *n* = 2 and 250 µm calibrated microparticles *n* = 2), they showed the slow-down of inflammation progression after embolization on MRI, angiographic, and histologic analyses [23].

The Bonar score, and, in particular, its vascular subclass, has shown discordances and limitations regarding correlation with therapeutic responses [24,25]. Our results add to these limitations, as there was no difference in the vascularity subclass of the Bonar score after embolization, although the number of neovessels was significantly different between controls and embolized piglets. This score does not seem to be optimal in tendinopathy evaluation and another tool to evaluate neoangiogenesis in a more adequate manner, through anatomical imaging (DSA, MRI) or functional imaging (radiotracer), should be considered.

### 4.2. Safety

Persistent cutaneous ischemia was reported in our study in piglets injected with small microspheres and IMP/CS, but not in those injected with larger microspheres (>100 µm). Although the “pruning” technique was used and emboli were administered slowly and in small amounts at a time, off-target embolization reports suggest that the emboli are passing through the capillary network. The use of skin ice packs near the embolization area has been reported with definitive microspheres to induce the vasoconstriction of collaterals and skin vessels and decrease the risk of off-target embolization [17]. Given the size of the IMP/CS crystals, it is logical that there would be non-target migration but the soluble nature of the crystals makes it transient. Nevertheless, we found residual crystals in a skin territory one week after embolization as reported in a frozen shoulder mouse model [22]. In contrast, a study evaluating rat kidneys 48 h after embolization found no residual crystals in the renal arteries [26]. A Japanese team recently compared IMP/CS embolization (six knees) versus resorbing gel foam (six knees) in a porcine model of knee arthritis. No complication and no off-target embolization agents were found 72 h after the procedure [27].

### 4.3. In Vitro Analysis

As previously reported [26], almost all IMP/CS particles were found to be smaller than 40 µm. The particle size distribution varied slightly between the emulsion with contrast and NaCl, as it seems to vary with the amount of contrast or with the time after suspension [26]. The transient nature of IMP/CS embolization is related to its solubility, which probably varies with its concentration and flow in the artery. In poorly vascularized terminal territories, or in case of over-embolization (too many injected particles), it is likely that the resorption time would increase. This could expose patients to ischemic complications from definitive microspheres [13].Furthermore, optimization of the IMP/CS embolization protocol should be performed, such as diluting the IMP/CS in contrast medium as well as the amount and duration of IMP/CS injection, to ensure patient safety and homogenous practices. Indeed, the protocol is likely to be different depending on the embolization indications and the arteries treated, which will have to be clarified for future studies. If the safety of the IMP/CS comes from its transient ischemic character, it is to be noted that in the clinical uses reported, digestive embolization [28] and osteoarticular pathologies [11,12,18], the arterial network is rich in collaterals and the risk of off-target embolization is higher. On the contrary, although less aggressive than definitive microparticles and despite its transient character, in terminal vascularization, such as the kidney, IMP/CS seems to be responsible for ischemic complications at 48 h [26]. Another hypothesis could be that the endothelium of neovessels may be different from that of normal vessels and are thus more sensitive to even a transient embolic agent and more at risk of thrombosis, whereas healthy vessels would have a greater tendency to recanalization [29].

### 4.4. Limitations

Our study has some limitations, among which is the small number of piglets. In addition, neovessel evaluation on angiography was performed visually. The use of software to count neovessels induced by collagenase injection and destroyed after embolization may strengthen the results and allow a more objective and reproducible evaluation. Lastly, this piglet model injected with collagenase did not show any clinical signs of tendinopathy [16]. The effect of embolization on pain and the walking score was not assessed. It would be interesting to study the clinical effects of these different embolization agents in other tendinopathy models.

## 5. Conclusions

Embolization in this porcine model with induced patellar tendinopathy was technically feasible with different embolization agents and allowed for assessing their safety and efficacy on neovessel destruction. The use of particles smaller than 100 µm seems to be associated with more complications than larger particles.

## Figures and Tables

**Figure 1 jpm-12-01530-f001:**
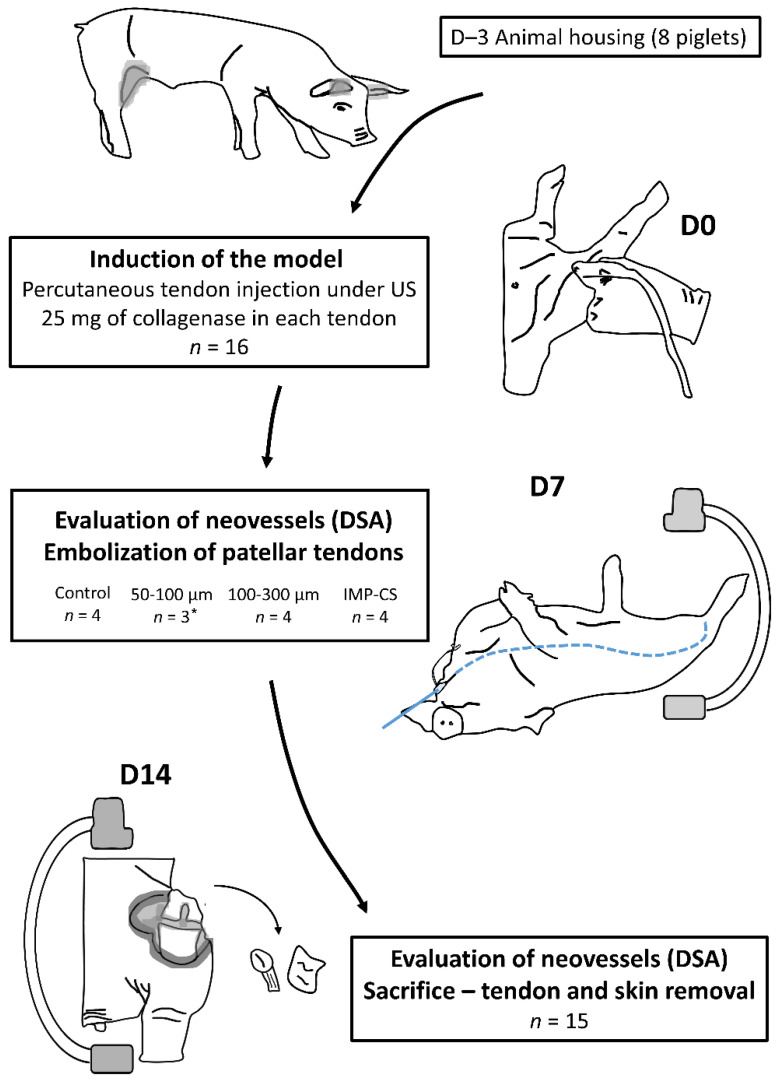
**Flowchart of the study.** US: ultrasound; D: day; DSA: digital subtraction angiography; *: one tendon was not treated due to femoral artery dissection on D7 angiography.

**Figure 2 jpm-12-01530-f002:**
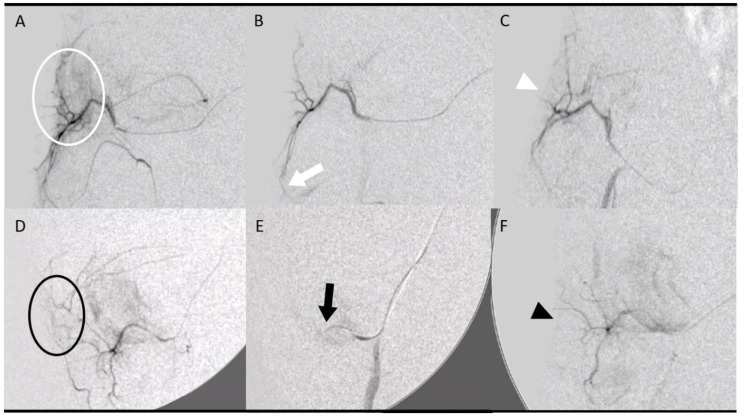
**Embolization procedure at D7 using various embolization agents: microspheres (A–C) and imipenem/cilastatin (D–F).** (**A**) Initial digital subtraction angiography (DSA) of a right geniculate artery showing neovascularization-related blush (white circle). (**B**) After embolization using 50–100 µm microspheres according to the “pruning” technique, DSA showed a “dead tree” aspect of the embolized area with a filling area by collaterals (white arrow). (**C**) DSA control at D14 showed no recurrence of neovascularization at the site of tendinopathy (white arrowhead). (**D**) The initial DSA of a right geniculate artery showing a blush related to neovascularization (black circle). (**E**) After embolization with an IMP/CS emulsion in an iodinated contrast medium, opacification revealed a truncated appearance of the geniculate artery (black arrow). (**F**) The DSA control at D14 showed repermeabilization of the native artery with no recurrence of neovessels (black arrowhead).

**Figure 3 jpm-12-01530-f003:**
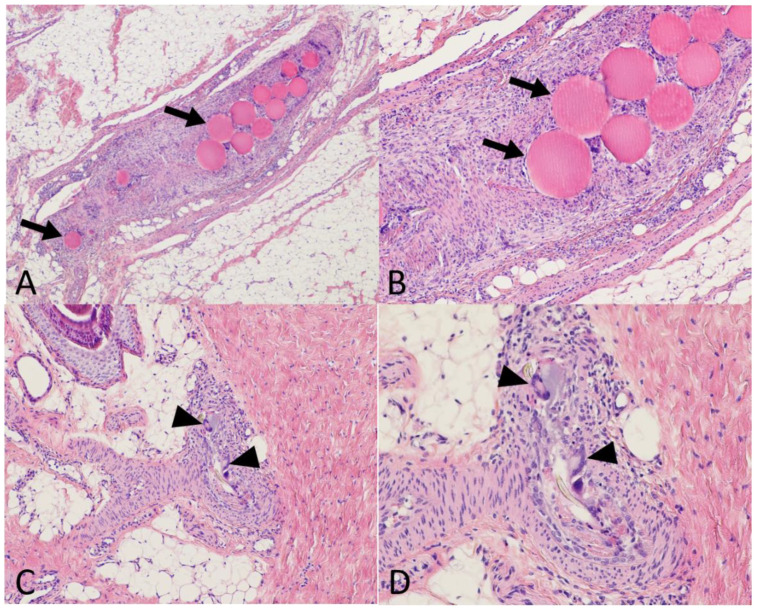
**Histological evaluation of the safety of embolization**. Using a standard stain (Hemathein Eosin Saffron) at ×40 (**A**), ×100 (**B**,**C**), and ×200 (**D**) magnification on tendon and skin samples taken 7 days after embolization. (**A**,**B**) Tendon treated by 50–100 µm microspheres. The peritendinous artery obliterated by the embolization material (arrows) with a granulomatous reaction. (**C**,**D**) Skin samples in the area of a transient livedo episode during embolization with Imipenem/Cilastatin (IMP/CS) emulsion. Dermohypodermal arteriole with granulomatous reaction with giant cells (black arrowheads) resorbing exogenous debris (IMP/CS crystals).

**Figure 4 jpm-12-01530-f004:**
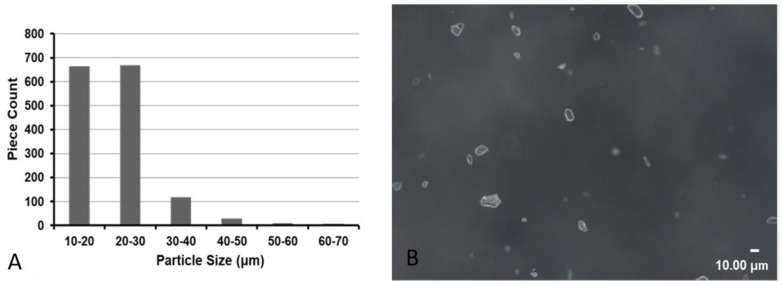
**In vitro exploration.** (**A**) Size distribution of 50 mg of imipenem/cilastatin (IMP/CS) particles in 1 mL of iodinated contrast dispersed during 10 s. (**B**) ×400 magnification of 50 mg/mL IMP/CS in iodinated contrast emulsion.

**Table 1 jpm-12-01530-t001:** Safety of various embolization agents.

Clinical Findings			Pathologic Findings at Day 14			
Groups	Immediate Transient Livedo	*p*-Value	Persistent Intracutaneous Ischemic Modifications	*p*-Value	Persistent Intracutaneous Embolization Agent	*p*-Value
Control (n = 4)	NA	0.76	NA	**0.02**	NA	0.48
50–100 (n = 3)	2		3		1	
100–300 (n = 4)	3		0		0	
IMP/CS (n = 4)	2		1		1	

IMP/CS: Imipenem/Cilastatin.

**Table 2 jpm-12-01530-t002:** Efficacy of various embolization agents.

Angiographic Neovascularization (Visual Evaluation)								
Groups	Day 7				Day 14			
	None	Mild	Severe	*p*-value	None	Mild	Severe	*p*-value
Controls (n = 4)	0	1	3	0.87	0	1	3	**0.04**
50–100 (n = 3)	0	1	2		2	1	0	
100–300 (n = 4)	0	1	3		2	2	0	
IMP/CS (n = 4)	0	2	2		3	1	0	
**Pathologic Findings on Day 14**								
Groups	Bonar global score	*p*-value	Bonar vascularity subclass	*p*-value	Neovessels/mm^2^	*p*-value	Intratendinous embolization agent	*p*-value
Controls (n = 4)	12 [10.5–13]	0.15	2.5 [1.8–3.0]	0.52	33 [29.5–36.8]	0.07	NA	**0.02**
50–100 (n = 3)	6 [6–7.5]		1 [1.0–1.5]		17 [16.5–20.5]		3	
100–300 (n = 4)	9 [6.5–11]		2 [0.8–3.0]		26 [24.8–27.8]		3	
IMP/CS (n = 4)	8.5 [6.8–9.5]		1 [1.0–1.3]		23 [18.3–27.3]		0	

IMP/CS: Imipenem/Cilastatin.

## Data Availability

Data will be accessible upon reasonable request to the corresponding author.

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
