# Peer review of "Feasibility of Neovessel Embolization in a Large Animal Model of Tendinopathy: Safety and Efficacy of Various Embolization Agents"

_jpm, 2022, doi:10.3390/jpm12091530_

Round 1
Reviewer 1 Report
The authors describe their experiences with embolization of patellar tendons in their own (neovascularization) tendinopathy piglet model. Comparison was made between no embolization (control), embo with particles (50-100µ vs 100-250µ) and embo with imipenem-cilastatin. They found that it was feasible (technical succes was >90%, 1 failure due to dissection) with angiographic response, but apparently less significant histological respons. They also reported cases of deeper penetration of embolization material, ie in the skin, in the small particle group and the imi group.
The manuscript is well readable and clear. The subject is pertinent, as embolization for treating pain syndromes in musculoskeletal indications are on the rise. However, I have the following comments about the manuscript:
1. As stated by the authors, only a (very) small number of subjects was included in the study. This is ok, as it concerns a feasibility "concept". However, the concluding remarks should be softened. Eg, based on the study data, it is not possible to state that smaller particles more likely cause complications (although in IR practice this can be true). Also, embolization techniques can never be standardized in such a small population, as also demonstrated by figure 2b and 2e, showing different embo endpoints.
2. Refs 3 and 4 are about OA and strictly not about tendinopathy. This should be refrased or change the refs.
3. White arrow in fig 2b demonstrates filling area by collaterals, not collaterals.
4. some minor typo's
5. why would using automated neovessel counting strength the results?
Author Response
Please see the attachment
Dear Editor and reviewers,
Please find enclosed our revised manuscript (jpm-1890479). Please find below a point-by-point answer to the reviewers’ comments and the changes made in the text as to take the comments into account.
Mainly, we have modified the Introduction section to add content and references, as suggested by one reviewer. We have also softened the conclusion section and modified the limitations section. Last, we have made some language corrections and have modified a paragraph of the Discussion section as to make it clearer.
We have also added an author (Noelia Sanchez-Ballester) initially forgotten by mistake, who participated in the in vitro analyses allowing the realization of this study.
We thank the reviewers for their time and relevant comments. The manuscript is now improved and we hope it will found suitable for publication in JPM.
Best regards
Reviewer 1:
Comment 1) As stated by the authors, only a (very) small number of subjects was included in the study. This is ok, as it concerns a feasibility "concept". However, the concluding remarks should be softened. Eg, based on the study data, it is not possible to state that smaller particles more likely cause complications (although in IR practice this can be true). Also, embolization techniques can never be standardized in such a small population, as also demonstrated by figure 2b and 2e, showing different embo endpoints.
Author response) Changed as suggested, both in the abstract and in the text. We have modified the Conclusion section and the In vitro analysis and Limitations sections (4.3 and 4.4) of the Discussion section.
Comment 2) Refs 3 and 4 are about OA and strictly not about tendinopathy. This should be refrased or change the refs.
Author response) Reference 3 has been changed and replaced with a more relevant reference. We have kept reference 4 and reworded the sentence.
Comment 3) White arrow in fig 2b demonstrates filling area by collaterals, not collaterals.
Author response) We have modified both the Figure and Figure legend as suggested by the reviewer. We also improved the visual of figure 2.
Comment 4) Some minor typo's.
Author response) The manuscript has been proofread to correct language and grammatical errors.
Comment 5) why would using automated neovessel counting strength the results?
Author response) A radiologist's visual assessment of the intensity of a vascular blush on angiography is subjective and subject to some intra- and inter-reader variations. The use of a software to evaluate the neovessel count would likely make the assessment more reproducible. We have reworded the sentence to be clearer on that point.

Reviewer 2 Report
Ghelfi et al. explored a study to investigate the feasibility of embolization with different embolization agents in a porcine model of patellar tendinopathy.
My comments is about the Introduction part, the intro is very scrace, not enought articles stressing the case, aor aorund the model.
About the us eof statistical element, I am sue the auyhors can reffre to the papers based on the statistival assesment. Please, do.
The limitatina nd conclusion section was not clear at all. I see the lack of any SI data.
Author Response
Please see the attachment
Dear Editor and reviewers,
Please find enclosed our revised manuscript (jpm-1890479). Please find below a point-by-point answer to the reviewers’ comments and the changes made in the text as to take the comments into account.
Mainly, we have modified the Introduction section to add content and references, as suggested by one reviewer. We have also softened the conclusion section and modified the limitations section. Last, we have made some language corrections and have modified a paragraph of the Discussion section as to make it clearer.
We have also added an author (Noelia Sanchez-Ballester) initially forgotten by mistake, who participated in the in vitro analyses allowing the realization of this study.
We thank the reviewers for their time and relevant comments. The manuscript is now improved and we hope it will found suitable for publication in JPM.
Best regards
Reviewer 2:
Comment 1) My comments is about the Introduction part, the intro is very scarce, not enough articles stressing the case, or around the model.
Author response) We have taken into account the reviewer’s comment and added elements and references in the Introduction section as to strengthen it and make it more consistent.
Comment 2) About the use of statistical element, I am sur the authors can reffer to the papers based on the statistical assessment. Please, do.
Author response) As requested we added the reference of the software used for analysis (Graph Pad Prism version 9.3 (GraphPad Software, CA, USA)).
Comment 3) The limitation and conclusion section was not clear at all.
Author response) We have taken into account the reviewer’s comment and modified both the Limitations and Conclusion sections.
Comment 4) I see the lack of any SI data.
Author response) We are sorry we are not sure to understand the reviewer’s comment. What does he/she mean by SI data ?

Round 2
Reviewer 2 Report
The authors address stated remarks.